# Impact of Triple Inhaler Therapy on COPD Patients with Non-Small Cell Lung Cancer After Radical Surgery: A Single-Centre Retrospective Analysis

**DOI:** 10.3390/jcm14010249

**Published:** 2025-01-03

**Authors:** Francesco Rocco Bertuccio, Vito D’Agnano, Simone Cordoni, Mitela Tafa, Cristina Novy, Nicola Baio, Klodjana Mucaj, Chandra Bortolotto, Giulio Melloni, Andrea Bianco, Angelo Guido Corsico, Fabio Perrotta, Giulia Maria Stella

**Affiliations:** 1Department of Internal Medicine and Medical Therapeutics, University of Pavia Medical School, 27100 Pavia, Italy; francesco.bertuccio01@gmail.com (F.R.B.); simone.cordoni01@universitadipavia.it (S.C.); mitela.tafa01@universitadipavia.it (M.T.); cristina.novy01@universitadipavia.it (C.N.); nicola.baio01@universitadipavia.it (N.B.); klodjana.mucaj01@universitadipavia.it (K.M.); g.melloni@smatteo.pv.it (G.M.); angelo.corsico@unipv.it (A.G.C.); 2Unit of Respiratory Disease, Cardiothoracic and Vascular Department, IRCCS Policlinico San Matteo, 27100 Pavia, Italy; 3Department of Translational Medical Sciences, University of Campania L. Vanvitelli, 80131 Naples, Italy; vito.dagnano@studenti.unicampania.it (V.D.); andrea.bianco@unicampania.it (A.B.); fabio.perrotta@unicampania.it (F.P.); 4U.O.C. Clinica Pneumologica L. Vanvitelli, Monaldi Hospital, A.O. dei Colli, 80131 Naples, Italy; 5Diagnostic Imaging and Radiotherapy Unit, Department of Clinical, Surgical, Diagnostic, and Pediatric Sciences, University of Pavia Medical School, 27100 Pavia, Italy; chandra.bortolotto@unipv.it; 6Radiology Institute, Fondazione Istituto di Ricovero e Cura a Carattere Scientifico (IRCCS) Policlinico San Matteo, 27100 Pavia, Italy; 7Unit of Thoracic Surgery, Cardiothoracic and Vascular Department, IRCCS Policlinico San Matteo, 27100 Pavia, Italy

**Keywords:** lung cancer, COPD, Thoracic surgery, triple inhaler therapy

## Abstract

**Background:** Chronic obstructive pulmonary disease (COPD) is among the most relevant comorbidity associated with lung cancer. The advent of innovative triple treatment approaches for COPD has significantly improved patients’ quality of life and outcomes. Few data are available regarding the impact of triple inhaler therapy on patients featuring COPD and lung cancer. **Methods:** We retrospectively evaluated the impact of triple inhale bronchodilators in a cohort of 56 patients with treated COPD who underwent lung surgery for primary cancer. **Results:** Triple bronchodilation can help to relieve the symptoms of the disease and improve lung function, allowing people with lung cancer to reduce the risk of serious exacerbations and improve their quality of life. **Conclusions:** Within the limits of the study, it should be underlined that bronchodilators can effectively affect the outcome and performance status after thoracic surgery.

## 1. Introduction

### 1.1. Lung Cancer and COPD

Lung cancer is among the major causes of cancer-related deaths worldwide, with only a 16% 5-year survival rate [1,2]. Chronic obstructive pulmonary disease (COPD) consists of a progressive deterioration of lung function over time and has a huge impact on a patient’s quality of life. It affects 50% of smokers and is one of the most common causes of death worldwide [3,4,5]. Cigarette smoking has a primary role as the causative agent in both COPD and lung cancer, and specific molecular patterns link the two diseases beyond a common etiology. Thus, COPD behaves as an independent risk factor for lung carcinoma, which occurs in this population five times more than in subjects with normal lung function [6,7]. According to current guidelines, surgery is the standard of care for the early-stage tumor when feasible [8]. Indeed, patients with severe COPD, although carrying resectable lung cancer, generally cannot undergo surgery due to the high risk of complications. Therefore, spirometry is an essential test before surgery. Several studies have reported that preoperative treatment with long-acting muscarinic antagonist (LAMA), long-acting β_2_-agonist (LABA), and/or inhaled corticosteroids (ICS) reduces the incidence of postoperative complications [9,10,11]. In postoperative outcomes, few studies describe patients with moderate-to-severe COPD, highlighting a higher prevalence of complications [12,13]. Moreover, few works have explored the effect of the severity of COPD on surgical outcomes in a small group of patients with mild COPD [14,15]. Functional (the degree of airway obstruction) and clinical (acute flares of disease) parameters are known to be determinants of survival outcomes in individuals affected by chronic obstructive pulmonary disease (COPD) [16,17]. Thus, it is concievable that acting on these two variables should also be effective in COPD patients referred to surgery for lung cancer resection. Until now, while treatment with ICS decreases mortality in COPD, only a small amount of data have been available regarding the potential role of COPD treatment in the survival and quality of life of patients with resected lung tumors [18,19].

### 1.2. The Role of Bronchodilators

Airway-targeted therapy, such as long-acting muscarinic antagonists (LAMA), long-acting beta-agonists (LABA), corticosteroids delivered via inhalation (ICS), and combination protocols, can improve lung function and quality of life and reduce exacerbations in COPD; it is thus reasonable to consider bronchodilators as fundamental in the therapy of COPD and concurrent lung cancer. In 2022, Hyunji Jo et al. reported that COPD treatment was associated with enhanced survival time in patients with advanced non-small cell lung cancer (NSCLC) and COPD and that a proper therapeutic approach to COPD should be a pivotal treatment for patients with advanced NSCLC [20]. A more recent study highlighted the potential use of bronchodilators as a useful strategy for lung cancer patients with breathing difficulties [21]. Beta-agonists and anticholinergics represent two main pharmacological classes of bronchodilators approved for COPD treatment. Beta-agonists induce dilation of the muscles of the airways by stimulating the beta-receptors. Anticholinergics act on muscarinic type 3 receptors (M3) and block acetylcholine, a neurotransmitter that induces smooth muscle fiber contraction. Both types of bronchodilators are known to mitigate dyspnoea in both COPD and lung cancer, and they are also used in a combinatorial approach with other medications, such as corticosteroids, to provide additional relief [22]. Overall, these molecules are often delivered through an inhaler, such as pressurized metered-dose inhalers (pMDIs), dry powder inhalers (DPI), soft mist inhalers (SMIs), and nebulizers. Every device is characterized by distinctive benefits and challenges, mainly regarding formulation requirements and patient usability. These are extremely relevant issues and the extent to which inhaler device user errors have been extensively prevented [23,24,25]. While better patient education can help with some mistakes, it is mandatory to develop innovative technologies that allow improvement in their usability to effectively impact patients’ outcomes [26]. Among the commercially available inhalers, MDIs are the most diffusely utilized [27]. MDI formulations can be categorized as solution- or suspension-based, depending on whether the drug is in a solution or solid form. Moreover, every inhaler (MDI, DPI, SMI, or nebulizer) has its own formulation and challenges concerning achieving dose consistency. This aspect is of great relevance mainly for those pharmaceutical products containing more than one drug, which are delivered in mono and dual formulations [28,29,30,31]. For suspension-based MDIs, a potential limitation is inadequate physical and motion stability, which can result in significant variation in the dose of the drug delivered each administration. To address this criticism, a co-suspension delivery technology has been designed. It consists of an innovative MDI formulation that suspends micronized drug crystals with spray-dried phospholipid excipient porous particles in hydrofluoroalkane (HFA) propellant [32]. In 2017, Doty and colleagues demonstrated that co-suspension delivery technology is able to enable the manufacture of an MDI suspension that is stable, consistent, and readily distributed when combined with one or more kinds of medication crystals.

This finding formulation enables the pharmaceutical development of drug products that overcome the most relevant MDI-related delivery consistency challenges, among which are handling errors. Glycopyrrolate/formoterol fumarate (GFF) MDI, a fixed-dose combination (FDC) of an anticholinergic and beta-agonist, glycopyrrolate, and formoterol fumarate, created using co-suspension delivery technology, exhibits reliable drug administration and aerosol performance under simulated handling mistakes and a range of conventional test settings [26]. Moreover, novel inhalers based on co-suspension metered formulations have been developed to ideally optimize the delivery of the most potent compound [30]. Small airway disease (SAD) has gained prominence recently as a significant pathophysiological characteristic of chronic obstructive lung disease and, as a result, is a primary cause of obstruction and symptoms [33]. Therefore, it is now considered a critical pharmacotherapy target. In 2023, Usmani et al. [34]. published a study that tested lung deposition of two ICS/LABA/LAMA single-inhaler triple therapies using in silico functional respiratory imaging (FRI). Real-world inhalation profiles that mimic daily use, where ideal inhalation may be disrupted, were used to evaluate deposition. Twenty individuals with moderate-to-very-severe COPD had their airways modeled in three dimensions. Using in silico FRI based on in vitro aerodynamic particle size distributions of each device, total, central, and regional small airways deposition as a percentage of the delivered dose of fluticasone furoate/umeclidinium/vilanterol (FF/UM/VI) 100/62.5/25 µg and budesonide/glycopyrronium/formoterol fumarate dihydrate (BGF) 160/7.2/5 µg per actuation were assessed.

The analysis’s primary finding was that, in comparison to FF/UM/VI, BGF was linked to higher total, central, and small airway deposition for all components. Crucially, only a tiny portion of the ICS from FF/UM/VI reached the small airways, resulting in a roughly five-fold variation in small airway deposition for the ICS components utilizing an equal inhalation profile. Despite certain study limitations, more investigation is required to determine whether improved BGF delivery results in clinical advantages such as higher mortality reductions [35].

Based on these observations, in the present work, we evaluated the impact of bronchodilators in COPD patients affected by early-stage lung cancer who underwent surgical treatment. A large amount of already available literature [36,37,38,39] focuses on the effects of bronchodilator therapy combined with smoking cessation, not on the operability of early-stage lung cancer. Herein, we aim to analyze if and how modulation of bronchodilation can affect the outcome of patients who undergo lung resection for lung cancer. This issue is worth evaluating since the pool of COPD patients with potentially resectable lung tumors is set to grow significantly in the future due to the advent of mini-invasive and robotic surgical techniques and the rising evidence that lung segmentectomy is a more effective approach than lobectomy in case of early-stage disease [40,41,42]. Proper COPD management is thus mandatory in this specific cancer patient population, and it requires tailored approaches and not a mere application of COPD recommendations and guidelines [43,44,45]. From this perspective, we did not report the functional spirometry data before and after surgery; we only reported the baseline COPD stage with respect to the bronchodilator applied.

## 2. Methods

We retrospectively investigated patients diagnosed with early-stage NSCLC and who underwent surgical treatment between January 2019 and December 2022 at the IRCCS Policlinico San Matteo Hospital. Among them, patients previously diagnosed with COPD or patients who had already received inhaler therapy were included in the present study. The post-bronchodilator Tiffenau index < 0.7 was applied to define COPD. Exclusion criteria were applied to individuals who received chemotherapy and or radiotherapy and/or did not have a histological diagnosis. Exhaustive demographic and clinical data are detailed in Table 1.

### 2.1. Data Collection and Assessment

Baseline clinical features including age, sex, smoking history, histological type, lung cancer stage, cancer treatment surgery, COPD treatment regimen, number of COPD exacerbations in the past year, lung function parameters (FVC, FEV1, DLCO, etc.), death before or after 6 months from the surgery were collected retrospectively using medical records from outpatient’s follow-up. Spirometry was used to detect COPD and assess the degree of airflow restriction during a year prior to and following the diagnosis of lung cancer.

COPD exacerbation events were reported together with the MMRC dyspnea scale if detected at outpatient follow-up or emergency visits.

### 2.2. Statistical Analysis

Basic descriptive analysis was assessed in the study population. The Student’s *t*-test for independent variables was used to compare the mean value ± the standard deviation (SD), which was used to express the continuous variables. A *p*-value < 0.05 was deemed statistically significant. The JMP partition algorithm (JMP-Statistical Discoveries; from SAS, website at www.jmp.com) was then used to evaluate the complete dataset. This technique can find all potential subdivisions of the best response/outcome predictors [46].

### 2.3. Outcomes

To clearly understand the impact of bronchodilators in this special population, we investigated the association between the frequency of COPD exacerbation, lung function and DLCO decline and the measurement of the MMRC dyspnea scale during outpatient follow-up.

## 3. Results

Between January 2019 and December 2022, fifty-six patients with COPD underwent lung cancer surgery (pulmonary resection of the hylo-mediastinal lymph nodes at the IRCCS Policlinico San Matteo Hospital and retrospectively analyzed. Thirty-two (48.48%) patients were male, and the median age at cancer diagnosis was 71.8 years. Four patients (7.14%) had a severe airflow obstruction (<50% predicted FEV 1) according to the Global Initiative for chronic Lung Disease (GOLD, website at https://goldcopd.org/gold-spirometry-guide/, accessed on 1 December 2024), while in the vast majority, a mild-moderate form was identified. Concerning smoking habits, all patients except two were former smokers. All patients involved in the analysis were diagnosed with COPD (defined as a post-bronchodilator FEV1/FVC ratio < 0.70). In our cohort, 39 patients were diagnosed with adenocarcinoma, while 17 had squamous cell carcinoma. Each case was evaluated by the institutional lung cancer board and staged as I and II based on the TNM 8 edition. The disease stage was then confirmed on postoperative analyses. As to surgical treatment, lobectomies were performed in 42 (75%) patients, atypical wedge resection in 11 (19.64%) and anatomical segmentectomy in 3 (5.37%) patients. Surgery was performed with radical intent, and in none of the evaluated cases chemotherapy and/or radiotherapy were required in an adjuvant setting. The surgeon’s judgment and the patient’s clinical features were used to customize the management of chest drainage and discharge planning for each treatment, not influenced by the type of surgery performed. No prolonged air leaks were observed in the cohort evaluated. Patients were chronically treated with bronchodilators, and they were restarted very soon, a few hours after surgery.

In order to better categorize which patients could benefit from a triple inhaler therapy [47,48,49,50], we investigated the eosinophil count in the complete blood count during the year of surgery. A total of 14 of 56 in our cohort had an eosinophil count >300 × 10^6^, 17 between 100 and 300 × 10^6^; despite that data, only 4 patients with higher eosinophil count (>300 × 10^6^) were on triple inhaler therapy. This finding reflects the fact that inhalation therapy could be optimized to have greater benefits in terms of reduction of symptoms and flare-ups. A promising cohort for tailored treatment is subgroups of lung cancer patients with high blood eosinophil levels. Due to their twin functions of fostering anti-tumor immunity and causing chronic inflammation, eosinophils—important mediators of inflammation—may have an impact on tumor microenvironments [51,52,53]. A unique strategy for improving patient outcomes is the combination of triple inhalation therapy with conventional lung cancer treatments of chemotherapy, surgery, and radiation. Up to 50% of individuals with lung cancer also have other obstructive airway illnesses, such as chronic obstructive pulmonary disease (COPD). Triple therapy is a potentially useful supplementary treatment in the context of lung cancer since it has demonstrated effectiveness in lowering exacerbations, improving lung function, and improving quality of life in people with COPD [54]. According to the newest research, the ICS components of triple therapy may have immunomodulatory and anti-inflammatory effects that work in concert with immunotherapy to improve anti-tumor responses and may lessen the inflammatory adverse effects of radiation or chemotherapy. Additionally, the components of LABA and LAMA enhance airway dynamics, which may help improve oxygenation and lessen pulmonary side effects during cancer therapies, and better respiratory function may lower perioperative risks and increase surgical readiness. Notably, the cohort enrolled in the present study, based on the disease stage, did not undergo treatments other than surgery. Moreover, the study design is not the proper one for analyzing molecular interactions between different drugs.

However, it is clearly evident that future research should explore the interplay between triple inhaler therapy and lung cancer treatments in randomized controlled trials [39,55,56]. Key areas include evaluating the effects on respiratory outcomes, therapy tolerability, and overall survival. This multidisciplinary approach may offer a pathway to integrating chronic respiratory disease management with oncologic care, addressing a critical overlap in lung cancer patients with obstructive airway disease. Currently, patients involved in the study are under clinical and instrumental follow-up in our department. Mortality in less than 6 months from the surgical treatment was 0% in our cohort, with five total deaths up to now. In our population, we recognized 13 (23.21%) patients that used triple inhaler therapy (four open triple therapy), 10 dual, and 29 monotherapies with a single bronchodilator. Concerning quality of life, we used the the MMRC dyspnea scale and the reported complaints by patients in outpatient follow-up as a measure of evaluation. Interestingly, 4 out of 14 patients reporting acute mild/moderate or severe exacerbation in the last year of observation assumed triple therapy. Among them, two patients were switched from dual to triple inhaler therapy for the persistence of symptoms and frequent acute mild exacerbations, whereas one patient was switched from monotherapy to dual bronchodilators with partial resolution of daily dyspnea. By evaluating the median and standard deviation of the distribution by applying the Student’s *t*-test, it seems that the aerosphere technology is associated with a lower number of exacerbations, one of the most relevant parameters that can impact patients’ survival, mainly after surgery for lung cancer (Figure 1). We did not perform the analysis with respect to lung surgery since this variable is related to lung cancer stage and not COPD severity.

## 4. Discussion

Our focus is based on the analysis of the impact of bronchodilators on patients affected by early-stage lung cancer who underwent surgical treatment.

In terms of survival, recent large-scale randomized trials have shown that ICS-containing combination therapy is better than dual bronchodilator therapy with LAMA-LABA [18,19]. Interestingly, according to a nationwide study of COPD patients, ICS users in Korea seem to have a reduced incidence of lung cancer than nonusers [39]. It is theorized that ICS lowers the risk of lung cancer, a form of chronic inflammation, by reducing the incidence of acute exacerbations in people with COPD through anti-inflammatory activity. In contrast, ICS was demonstrated to increase the survival of patients with advanced non-small cell lung cancer in another trial. According to the authors’ premise, the anti-inflammatory properties of ICS prevent the spread of cancer [18,19].

It is interesting to note that ICS users in Korea appear to have a lower incidence of lung cancer than nonusers, based on a nationwide study of COPD patients [57]. It is theorized that ICS lowers the risk of lung cancer, a form of chronic inflammation, by reducing the incidence of acute exacerbations in individuals with COPD through an anti-inflammatory activity. In contrast, ICS was demonstrated to increase the survival of patients with advanced non-small cell lung cancer in another trial. The authors postulated that ICS’s anti-inflammatory properties prevent the spread of cancer [20]. Moreover, we have shown that respiratory function parameters are among the most relevant predictors of outcomes in lung cancer patients who undergo surgical treatment [58]. It is relevant to note that bronchodilators, either alone or in combination, can also successfully prevent COPD flare-ups. Exacerbations are seen as significant occurrences in the clinical progression of COPD, and current treatment strategies or guidelines emphasize preventing exacerbations as a crucial therapeutic objective and pertinent outcome measure.

To clearly understand the impact of bronchodilators in this special population, we investigated the association between the frequency of COPD exacerbation, lung function and DLCO decline and the measurement of the MMRC dyspnea scale during outpatient follow-up. Exacerbations are regarded as significant occurrences in COPD patients’ clinical progression, especially in our subset population affected by lung cancer; the prevention of exacerbations has a pivotal role, especially considering long-term outcomes and mortality. Overall, studies suggest that the co-suspension delivery technology, which is designed to overcome suspension heterogeneity, reduce inappropriate drug–drug interaction and minimize handling errors, may offer an effective approach to act on both central and peripheral airways. In terms of lung volume analysis, postoperative pulmonary function is difficult to predict accurately because it changes from the time of the operation and is also affected by various factors, such as the type of surgical intervention. Our approach highlighted some unexpected findings: from our work, it is clear that many patients (37 out of 56) did not refer to any exacerbation of COPD over the period of observation. Indeed, of the 13 patients on triple inhaler therapy, 3 patients had a mild exacerbation and 2 serious events with hospitalization. It is worth noting that severe airflow obstruction was detected among them before the surgical treatment; consequently, they were affected by a more advanced disease. Despite that, in our population, the reduction of both FEV1 and DLCO before and after surgery was deemed to be physiological due to the intervention, and we did not register any further unexpected decline. Thus, a triple inhaler treatment combined with the aerosphere was found to stabilize the FEV1 and DLCO values in four out of the 13 patients’ analyses. Our work’s primary limitation is that it is a single-center study; consequently, the statistical sample we examined is limited. It would, therefore, be useful to do further analysis (maybe a multicentric analysis) and validation for the relevant clinical, diagnostical and prognostic implications that the findings we have found deserve. There has been debate on the possible ways that bronchodilators might prevent exacerbations.

In addition to indirect mechanisms (better secretion clearance through improved airway patency) and the anti-inflammatory qualities of bronchodilators (reduced sputum production, cytokine release), these may include direct effects on airflow, reduced hyperinflation, which improves respiratory mechanics and raises thresholds for the onset of symptoms. In addition, Cazzola et al. published an analysis in 2022 involving a real-world population of patients with COPD and a history of exacerbations; compared to dual bronchodilation, the commencement of triple therapy was linked to a higher reduction in the risk of acute respiratory episodes, future exacerbations, and treatment failure [59]. Among currently available strategies to prevent these events, maintenance therapy with bronchodilators is of great relevance.

The potential mechanisms by which inhaler therapy could prevent acute events have been discussed for many years. The main reported hypothesis refers to the effects of direct impact on airflow, reduction of hyperinflation with concurrent improved respiratory mechanics, improved clearance of airway secretions and anti-inflammatory properties (especially of ICS). As previously mentioned, research employing functional respiratory imaging and gamma scintigraphy has shown that GFF MDI is efficiently deposited in the central and peripheral airways and offers clinically significant improvements in airway volume and resistance throughout the lung. Treatment with fixed-dose triple therapy reduced the incidence of acute respiratory events more than dual bronchodilation did for individuals with higher blood eosinophil counts and a history of past exacerbations. Nonetheless, there was no discernible variation in the advantages of triple therapy according to risk category or GOLD severity [39]. As discussed above, it should also be noted that eosinophils should be implicated in responses to cancer therapy. Better responses to immunotherapy, especially immune checkpoint inhibitors (ICIs), have been linked to elevated eosinophil levels, which may indicate a predictive biomarker role [60,61,62,63]. ICS-containing regimens may be beneficial for patients with eosinophilic lung cancer when used in conjunction with inhalation treatment. ICS could lower eosinophilic inflammation, which is important for people who also have asthma or eosinophilic COPD [64,65]. This could lead to improved symptom control, reduced exacerbation frequency, and better overall respiratory status, potentially enabling greater tolerance for systemic lung cancer therapies.

Additionally, in situations where immunotherapy and eosinophil-driven inflammation work in concert, ICS may alter immunological responses, increasing the effectiveness of ICIs [62,66,67]. It is necessary to exercise caution, though, because ICS can weaken innate immune responses, which could make immunocompromised cancer patients more susceptible to infections. This emphasizes how important it is to carefully choose patients when thinking about inhalation treatments for eosinophilic lung cancer. Future studies should examine whether ICS, when used in triple inhaler therapy, enhances clinical outcomes in this subgroup, including overall survival, therapy tolerability, and symptom load.

Furthermore, eosinophilic status may directly tailor strategies for combining oncologic and pulmonary care. In conclusion, within the limitation of the cohort analyzed, the subpopulation with improved clinical outcome (defined as reduced COPD exacerbation event number and severity as well as better quality of life after thoracic surgery for lung cancer) by using triple bronchodilation. The co-suspension delivery technology assuring drugs homogenously dissolve within the airways deserves specific investigation and is associated with the most promising performances. Overall, it should be underlined that the present study’s design and results do not allow any mechanistic implication and conclusion regarding the role of triple bronchodilation in the oncologic features of each case evaluated. Dedicated studies exploring the interaction between triple inhaler therapy and other modalities, such as chemotherapy, immunotherapy, or targeted therapies, would provide valuable insights and deserve significant interest in the near future.

## 5. Conclusions

Bronchodilators can effectively impact the outcome of patients affected by lung cancer after surgery. Overall, they can help to relieve the symptoms of the disease and improve lung function, allowing people with lung cancer to reduce the risk of serious exacerbations and improve their quality of life. Research innovation is thus aimed at developing inhalers that can ensure effective drug doses once they reach lung airways, which is the key issue in determining their therapeutic effect. The increasingly frequent application of minimally invasive techniques in thoracic surgery has allowed a significant increase in the number of patients who can undergo parenchymal resection after lung cancer diagnosis [68]. Expanding the cohort (e.g., conducting a multi-center analysis with a larger sample size) would enhance the reliability and generalizability of the results. Thus, the impact of triple airway dilation in this specific lung cancer population will deserve further and wider studies.

## Figures and Tables

**Figure 1 jcm-14-00249-f001:**
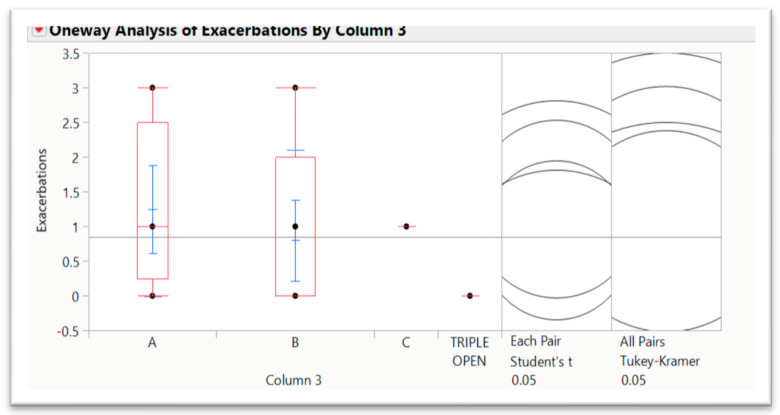
Distribution of COPD exacerbations based on the different triple inhaler therapies analyzed (A, B, C and triple open). Although non-stastically significant, the B therapy (corresponding to aerosphere technology) is associated with better performances.

**Table 1 jcm-14-00249-t001:** Demographic and clinical characteristics of the cohort analyzed. Yrs—years, SCC—squamous cell carcinoma; ADC—adenocarcinoma; pre-op—preoperative.

Variable	Number
Gender	
Male	32
Female	24
Smoking habit	
Never	3
Current/past	53
Median age at cancer diagnosis (yrs)	71.8
Histotype	
SCC	17
Non-SCC/ADC	39
Surgery	
Lobectomy	42
Segmentectomy	14
TNM stage	
I	32
II	24
Respiratory Function	
Pre-op FEV1 (%)	73
Pre-op DLCO (%)	69
Death after 12 months	1
Bronchodilation strategy	
Triple	13
Open triple	4
Dual	10
Mono	29
Exacerbation in the first year	14

## Data Availability

The data that support the findings of this study are available on request from the corresponding author.

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
