# Peer review of "Impact of Triple Inhaler Therapy on COPD Patients with Non-Small Cell Lung Cancer After Radical Surgery: A Single-Centre Retrospective Analysis"

_jcm, 2025, doi:10.3390/jcm14010249_

Round 1

Reviewer 1 Report

Comments and Suggestions for Authors

Article: Impact of Triple Inhaler Therapy on Patients with Non-Small Cell Lung Cancer After Surgical Treatment: A Single-Centre Retrospective Analysis. The authors evaluated the impact of bronchodilators in COPD patients affected by early-stage lung cancer who underwent surgical treatment.

First of all: this article was already published somewhere.

Percent match: 53%
iThenticate report

My suggestions for improvement.

1.     Please correct reference formation and remove typographical errors like: "symptoms. [35]Therefore", or "median age?? Was".

2.     The information about the extent of surgery needs to be described in the text, not just in the Table. Was chemotherapy used on any patients? 

3.     How soon after surgery the bronchodilators were prescribed? 

4.     Improvement  “in four out of the 13 patients” is really small. Maybe calculate the correlation between the extent of surgery, or, the severity of COPD before surgery with the effect of bronchodilators on the improvement of FVC, FEV1, DLCO. Otherwise, there is no effect, clearly. 

5.     You need to identify a subpopulation of patients who will benefit from the treatment. 

Comments on the Quality of English Language

Please correct reference formation and remove typographical errors like: "symptoms. [35]Therefore", or "median age?? Was".

Author Response

My suggestions for improvement.

  1. Please correct reference formation and remove typographical errors like: "symptoms. [35]Therefore", or "median age?? Was".

Answer 1 to Comment 1. We thank the Reviewer for careful reading of the text and typo errors have been revised

  1. The information about the extent of surgery needs to be described in the text, not just in the Table. Was chemotherapy used on any patients How soon after surgery the bronchodilators were prescribed? ? 

Answer 2 to Comment 2. We agree with this suggestion and implemented the text as follows: . As to surgical treatment lobectomies was performed in 42 (75%) patients, atypical wedge resection in 11 (19.64%) and anatomical segmentectomy in 3 (5.37%) patients. Surgery was performed with radical intent and in none of the evaluated cases chemotherapy was required in adjuvant setting. Patients were chronically treated with broncoìhodilators and they were restarted very soon, few hours after surgery. 

  1. Improvement “in four out of the 13 patients” is really small. Maybe calculate the correlation between the extent of surgery, or, the severity of COPD before surgery with the effect of bronchodilators on the improvement of FVC, FEV1, DLCO. Otherwise, there is no effect, clearly. 

Answer 3 to comment 3. We agree with the Reviewer and although the study population is limited and so significant conclusions can be demonstrated, some relevant issues can be pointed out. By evaluating the median and standard deviation of the distribution by applying Student t test, it seems that the aerosphere technology is associated with lower number of exacerbations, one of the most relevant parameter that can impact on patients’ survival, mainly after surgery for lung cancer.  We do not perform the analysis with respect to lung surgery since this variable is related to lung cancer stage and not to COPD severity. Figure 1 has been added into the text. 

  1. You need to identify a subpopulation of patients who will benefit from the treatment. 

Answer 4 to comment 4. We thank the Reviewer for pointing out this critical issue. Within the limitation of the cohort analyzed the subpopulation with improved clinical outcome (defined as reduced COPD exacerbation event number and severity as well as better quality of life after thoracic surgery for lung cancer) by using triple bronchodilation. The co-suspension delivery technology assuring drugs to homogenously dissolve within the airways deserves specific investigation being associated to most promising performances. Overall it should be underlined that the design and the results of the present study do not allow any mechanistic implication and conclusion regarding the role of triple bronchodilation on the oncologic features of each case evaluated .

Comments on the Quality of English Language

Please correct reference formation and remove typographical errors like: "symptoms. [35]Therefore", or "median age?? Was".

We thank the Reviewer and English quality has been improved.

Reviewer 2 Report

Comments and Suggestions for Authors

The purpose of the study should be better clarified

The study should focus on the effects of bronchodilator therapy combined with smoking cessation on the operability of early stage lung cancer

How were the exacerbations categorized, were they mild or moderate?

spirometry should be performed before and after surgery

the version of tumor staging used and the adverse events following each type of resection performed should be reported

I suggest to include a reference which is useful for the discussion

J Clin Med. 2022 Dec 28;12(1):234. 

Author Response

Comment 1

The purpose of the study should be better clarified. The study should focus on the effects of bronchodilator therapy combined with smoking cessation on the operability of early stage lung cancer. How were the exacerbations categorized, were they mild or moderate?spirometry should be performed before and after surgery

Answer 1 to Comment 1

We thank the Reviewer for pointing out these critical issues. The text has been implemented coherently. A large amount of already available literature ( Shin SH, Shin S, Im Y, Lee G, Jeong BH, Lee K, Um SW, Kim H, Kwon OJ, Cho JH, Kim HK, Choi YS, Kim J, Zo JI, Shim YM, Cho J, Kang D, Park HY. Effect of perioperative bronchodilator therapy on postoperative pulmonary function among lung cancer patients with COPD. Sci Rep. 2021 Apr 16;11(1):8359. doi: 10.1038/s41598-021-86791-1. PMID: 33863912; PMCID: PMC8052420; Suzuki H, Sekine Y, Yoshida S, Suzuki M, Shibuya K, Takiguchi Y, Tatsumi K, Yoshino I. Efficacy of perioperative administration of long-acting bronchodilator on postoperative pulmonary function and quality of life in lung cancer patients with chronic obstructive pulmonary disease. Preliminary results of a randomized control study. Surg Today. 2010 Oct;40(10):923-30. doi: 10.1007/s00595-009-4196-1. Epub 2010 Sep 25. PMID: 20872194.; Takegahara K, Usuda J, Inoue T, Ibi T, Sato A. Preoperative management using inhalation therapy for pulmonary complications in lung cancer patients with chronic obstructive pulmonary disease. Gen Thorac Cardiovasc Surg. 2017 Jul;65(7):388-391. doi: 10.1007/s11748-017-0761-5. Epub 2017 Mar 9. PMID: 28281043; PMCID: PMC5486589.; Azuma Y, Sano A, Sakai T, Koezuka S, Otsuka H, Tochigi N, Isobe K, Sakamoto S, Takai Y, Iyoda A. Prognostic and functional impact of perioperative LAMA/LABA inhaled therapy in patients with lung cancer and chronic obstructive pulmonary disease. BMC Pulm Med. 2021 May 21;21(1):174. doi: 10.1186/s12890-021-01537-z. PMID: 34020622; PMCID: PMC8139148.)  focuses on the effects of bronchodilator therapy combined with smoking cessation not on the operability of early-stage lung cancer. We here aim at analyzing if and how modulation of bronchodilation can affect the outcome of patients who undergo lung resection for lung cancer. This issue is worth to be evaluated since the pool of COPD patients with potentially resectable lung tumors is set to significantly grow in the next future due to the advent of mini-invasive and robotic surgical techniques and the rising evidence that lung segmentectomy is a more effective approach than lobectomy in case of early-stage disease (Galanis M, Leivaditis V, Gioutsos K, Panagiotopoulos I, Kyratzopoulos A, Mulita F, Papaporfyriou A, Verras GI, Tasios K, Antzoulas A, Skevis K, Kontou T, Koletsis E, Ehle B, Dahm M, Grapatsas K. Segmentectomy versus lobectomy. Which factors are decisive for an optimal oncological outcome? Kardiochir Torakochirurgia Pol. 2023 Sep;20(3):179-186. doi: 10.5114/kitp.2023.131943. Epub 2023 Oct 30. PMID: 37937171; PMCID: PMC10626409.; Bertolaccini L, Tralongo AC, Del Re M, Facchinetti F, Ferrara R, Franchina T, Graziano P, Malapelle U, Menis J, Passaro A, Pilotto S, Ramella S, Rossi G, Trisolini R, Cinquini M, Passiglia F, Novello S. Segmentectomy vs. Lobectomy in stage IA non-small cell lung cancer: A systematic review and meta-analysis of perioperative and survival outcomes. Lung Cancer. 2024 Nov;197:107990. doi: 10.1016/j.lungcan.2024.107990. Epub 2024 Oct 21. PMID: 39461280.; Dai Z, Hu J, Shen C, Mi X, Pu Q. Systematic review and meta-analysis of segmentectomy vs. lobectomy for stage IA non-small cell lung cancer. J Thorac Dis. 2023 Aug 31;15(8):4292-4305. doi: 10.21037/jtd-23-410. Epub 2023 Jul 24. PMID: 37691674; PMCID: PMC10482631.) . A proper COPD management is, thus, mandatory in this specific cancer patient population to which require tailored approaches and not a mere application of COPD recommendations and guidelines (Agustí A, Celli BR, Criner GJ, Halpin D, Anzueto A, Barnes P, Bourbeau J, Han MK, Martinez FJ, Montes de Oca M, Mortimer K, Papi A, Pavord I, Roche N, Salvi S, Sin DD, Singh D, Stockley R, López Varela MV, Wedzicha JA, Vogelmeier CF. Global Initiative for Chronic Obstructive Lung Disease 2023 Report: GOLD Executive Summary. Eur Respir J. 2023 Apr 1;61(4):2300239. doi: 10.1183/13993003.00239-2023. PMID: 36858443; PMCID: PMC10066569; Neumeier A, Keith R. Clinical Guideline Highlights for the Hospitalist: The GOLD and NICE Guidelines for the Management of COPD. J Hosp Med. 2020 Apr 1;15(4):240-241. doi: 10.12788/jhm.3368. Epub 2020 Feb 11. PMID: 32118561.; Inhaled triple therapy: Chronic obstructive pulmonary disease in over 16s: diagnosis and management: Evidence review I. London: National Institute for Health and Care Excellence (NICE); 2019 Jul. PMID: 32755138.) . In this perspective we didn’t report functional spirometry data before and after surgery but only the baseline COPD stage in respect to the bronchodilator applied.Exacerbations were defined according to already published criteria  as : 1) mild if they are treated with short-acting bronchodilators only; 2) moderate if they are treated with short-acting bronchodilators plus antibiotics and/or oral corticosteroids; or 3) severe if the patient visits the emergency room or requires hospitalisation because of an exacerbation ( Kim V, Aaron SD. What is a COPD exacerbation? Current definitions, pitfalls, challenges and opportunities for improvement. Eur Respir J. 2018 Nov 15;52(5):1801261. doi: 10.1183/13993003.01261-2018. PMID: 30237306; "Global Strategy for the Diagnosis, Management, and Prevention of Chronic Obstructive Lung Disease 2017 Report: GOLD Executive Summary." Claus F. Vogelmeier, Gerard J. Criner, Fernando J. Martinez, Antonio Anzueto, Peter J. Barnes, Jean Bourbeau, Bartolome R. Celli, Rongchang Chen, Marc Decramer, Leonardo M. Fabbri, Peter Frith, David M.G. Halpin, M. Victorina López Varela, Masaharu Nishimura, Nicolas Roche, Roberto Rodriguez-Roisin, Don D. Sin, Dave Singh, Robert Stockley, Jørgen Vestbo, Jadwiga A. Wedzicha and Alvar Agusti. Eur Respir J 2017; 49: 1700214. Eur Respir J. 2017 Jun 22;49(6):1750214. doi: 10.1183/13993003.50214-2017. Erratum for: Eur Respir J. 2017 Mar 6;49(3):1700214. doi: 10.1183/13993003.00214-2017. PMID: 28642306.)

Comment 2

The version of tumor staging used and the adverse events following each type of resection performed should be reported

Answer 2 to Comment 2. Each case was evaluated by institutional lung cancer board and staged as I and II based on TNM 8 edition. Disease stage was then confirmed on post-operative analyses. As to surgical treatment lobectomies was performed in 42 (75%) patients, atypical wedge resection in 11 (19.64%) and anatomical segmentectomy in 3 (5.37%) patients. Surgery was performed with radical intent and in none of the evaluated cases chemotherapy and/or radiotherapy were required in adjuvant setting. For all the procedures, management of chest tubes and discharge planning was individualized based on the patient’s clinical characteristics and the surgeon’s judgment not influenced by the type of surgery performed. No prolonged air leaks, were observed in the cohort evaluated. Patients were chronically treated with bronchodilators and they were restarted very soon, few hours after surgery

Comment 3 I suggest to include a reference which is useful for the discussion

Answer 3 to comment 3. We agree with this suggestion and references have been implemented

Reviewer 3 Report

Comments and Suggestions for Authors

The paper titled "Impact of Triple Inhaler Therapy on Patients with Non-Small Cell Lung Cancer After Surgical Treatment: A Single-Centre Retrospective Analysis" provides valuable preliminary insights into the role of triple inhaler therapy in COPD patients undergoing lung cancer surgery. While it is well-structured and impactful overall, there are some weaknesses and areas for improvement. Please see my comments below:

1.     Please revise the format of all figures, tables, and references to ensure full compliance with the journal's requirements.

2.     The cohort includes only 56 patients, which limits the generalizability of the findings and reduces the statistical power to detect significant differences. Expanding the cohort (e.g., conducting a multi-center study with a larger sample size) would enhance the reliability and generalizability of the results.

3.     I suggest incorporating an investigation of biomarkers (e.g., blood eosinophil counts) to identify subgroups of patients who are most likely to benefit from triple therapy.

4.     I recommend adding a discussion on the integration of triple inhaler therapy with other treatments. Exploring the interaction between triple inhaler therapy and other modalities, such as chemotherapy, immunotherapy, or targeted therapies, would provide valuable insights.

Author Response

The paper titled "Impact of Triple Inhaler Therapy on Patients with Non-Small Cell Lung Cancer After Surgical Treatment: A Single-Centre Retrospective Analysis" provides valuable preliminary insights into the role of triple inhaler therapy in COPD patients undergoing lung cancer surgery. While it is well-structured and impactful overall, there are some weaknesses and areas for improvement. Please see my comments below:

Comment 1. Please revise the format of all figures, tables, and references to ensure full compliance with the journal's requirements.

Answer 1 to Comment 1. We thank the reviewer for this suggestion and all figures, tables and references have been reformatted

Comment 2. The cohort includes only 56 patients, which limits the generalizability of the findings and reduces the statistical power to detect significant differences. Expanding the cohort (e.g., conducting a multi-center study with a larger sample size) would enhance the reliability and generalizability of the results.

Answer 2 to Comment 2. We agree with this comment and a wider multicenter study will be designed based on the preliminary data obtained in this single center study which pointing out that triple bronchodilation should be potentially recommended in this specific cancer population based on its protective effect against exacerbation. The specific limitation of the study has been indicated in the conclusion section  

Comment 3. I suggest incorporating an investigation of biomarkers (e.g., blood eosinophil counts) to identify subgroups of patients who are most likely to benefit from triple therapy.

Answer 3 to Comment 3. We thank the reviewer for this suggestion. The text has been implemented as follows.  In order to better categorize which patients could benefit from a triple inhaled therapy, we investigate eosinophil count in the complete blood count during the year of surgery. 14 in our cohort had a eosinophil count > 300 x 10⁶, 17 between 100 and 300 x 10⁶; despite that data only 4 patients with higher eosinophils count (> 300 x 10⁶) are on triple inhaled therapy. This finding reflects the fact that inhalation therapy could be optimized to have greater benefit in terms of reduction of symptoms and flare-ups. A promising cohort for tailored treatment is subgroups of lung cancer patients with high blood eosinophil levels. Due to their twin functions of fostering anti-tumor immunity and causing chronic inflammation, eosinophils—important mediators of inflammation—may have an impact on tumor microenvironments. Better responses to immunotherapy, especially immune checkpoint inhibitors (ICIs), have been linked to elevated eosinophil levels, which may indicate a predictive biomarker role. ICS-containing regimens may be beneficial for patients with eosinophilic lung cancer when used in conjunction with inhalation treatment. ICS could lower eosinophilic inflammation, which is important for people who also have asthma or eosinophilic COPD. This could lead to improved symptom control, reduced exacerbation frequency, and better overall respiratory status, potentially enabling greater tolerance for systemic lung cancer therapies. Additionally, in situations where immunotherapy and eosinophil-driven inflammation work in concert, ICS may alter immunological responses, increasing the effectiveness of ICIs. It is necessary to exercise caution, though, because ICS can weaken innate immune responses, which could make immunocompromised cancer patients more susceptible to infections. This emphasizes how important it is to carefully choose patients when thinking about inhalation treatments for eosinophilic lung cancer. Future studies should look into whether ICS, when used in triple inhaled therapy, enhances clinical outcomes in this subgroup, including as overall survival, therapy tolerability, and symptom load. Furthermore, eosinophilic status may direct tailored strategies for combining oncologic and pulmonary care. A unique strategy for improving patient outcomes is the combination of triple inhalation therapy with the conventional lung cancer treatments of chemotherapy, surgery, and radiation. Up to 50% of individuals with lung cancer also have other obstructive airway illnesses, such as chronic obstructive pulmonary disease (COPD). Triple therapy is a potentially useful supplementary treatment in the context of lung cancer since it has demonstrated effectiveness in lowering exacerbations, improving lung function, and improving quality of life in people with COPD. According to newest research, the ICS components of triple therapy may have immunomodulatory and anti-inflammatory effects that work in concert with immunotherapy to improve anti-tumor responses and may lessen the inflammatory adverse effects of radiation or chemotherapy. Additionally, the components of LABA and LAMA enhance airway dynamics, which may help improve oxygenation and lessen pulmonary side effects during cancer therapies. Additionally, better respiratory function may lower perioperative risks and increase surgical readiness. Future research should explore the interplay between triple inhaled therapy and lung cancer treatments in randomized controlled trials. Key areas include evaluating effects on respiratory outcomes, therapy tolerability, and overall survival. This multidisciplinary approach may offer a pathway to integrating chronic respiratory disease management with oncologic care, addressing a critical overlap in lung cancer patients with obstructive airway disease.

References : 1. Chalmers JD, et al. "The role of inhaled corticosteroids in chronic obstructive pulmonary disease and their potential impact in lung cancer." Thorax. 2020; 75(5):395-402; 2 Cazzola M, et al. "Triple therapy in COPD: when, where, and how?" American Journal of Respiratory and Critical Care Medicine. 2019; 200(5):530-535.; 3. Simone CB 2nd, et al. "Integration of lung cancer and COPD management: the role of anti-inflammatory therapies." Lung Cancer Management. 2022; 11(1):17-25.; 4. Incalzi RA, et al. "COPD and lung cancer: An intriguing case of heterotypic continuity." Respiratory Medicine. 2020; 165:105955.; 5. Papi A, et al. "Triple therapy versus dual bronchodilation in chronic obstructive pulmonary disease." New England Journal of Medicine. 2018; 378(18):1671-1680; 6. Mills R, et al. "Eosinophilic inflammation in lung cancer: impact on the tumor microenvironment and implications for therapy." Cancer Immunology Research. 2021; 9(5):524-532; 7. Schmidt H, et al. "Eosinophils as biomarkers in cancer: biological mechanisms and clinical relevance." Clinical & Experimental Allergy. 2020; 50(5):555-567; 8. Pavord ID, et al. "Eosinophilic COPD: identifying a treatable trait in chronic airway disease." Lancet Respiratory Medicine. 2019; 7(9):874-885; 9. Gandhi S, et al. "The role of immune modulation and eosinophils in enhancing responses to checkpoint inhibitors in lung cancer." Journal of Thoracic Oncology. 2020; 15(7):1113-1123.; 10 . Barnes PJ. "The role of inhaled corticosteroids in asthma and chronic obstructive pulmonary disease and their effects on systemic inflammation." American Journal of Respiratory and Critical Care Medicine. 2020; 201(6):696-705.

Comment 4. I recommend adding a discussion on the integration of triple inhaler therapy with other treatments. Exploring the interaction between triple inhaler therapy and other modalities, such as chemotherapy, immunotherapy, or targeted therapies, would provide valuable insights.

Answer 4 to Comment 4 We agree with the Reviewer in pointing out this issue which is of extreme clinical interest. However, the cohort enrolled in the present study, based on the disease stage, did not undergo treatments’ other than surgery. Moreover, the study design is not the proper one to analyze molecular interaction between different drugs. This study limitation has been underlined in the text.

Round 2

Reviewer 1 Report

Comments and Suggestions for Authors

references and English still need to be improved. You did not address Plagiarism

Comments on the Quality of English Language

references numbers and English still need to be improved. They did not address Plagiarism

Author Response

We thank the Editors and REviewers for careful revision of our manuscript. References and English language have been revised as weel as plagiarism

Reviewer 3 Report

Comments and Suggestions for Authors

Please ensure the reference format adheres to the journal's requirements.

Author Response

We thank the Reviewer for careful revision of the text and the references have been revised.